# Post-discharge mortality in suspected pediatric sepsis: Insights from rural and urban healthcare settings in Rwanda

Christian Umuhoza[1,2]*, Anneka Hooft[3], Cherri Zhang[4], Jessica Trawin[4], Cynthia Mfuranziza[5], Emmanuel Uwiragiye[6], Vuong Nguyen[4], Aaron Kornblith[3], Nathan Kenya Mugisha[7], J. Mark Ansermino[4], Matthew O. Wiens[4,7,8]*

**1** Department of Pediatrics, University Teaching Hospital of Kigali, Kigali, Rwanda, **2** Department of Pediatrics, University of Rwanda, Kigali, Rwanda, **3** Departments of Emergency Medicine and Pediatrics, University of California, San Francisco, California, United States of America, **4** Institute for Global Health, British Columbia Children's and Women's Hospital, Vancouver, Canada, **5** Rwanda Paediatric Association, Kigali, Rwanda, **6** Ruhengeri Regional Referral Hospital, Musanze, Rwanda, **7** Walimu, Kampala, Uganda, **8** Department of Anesthesiology, Pharmacology and Therapeutics, University of British Columbia, Vancouver, Canada

☙ These authors contributed equally to this work.
* christian.umuhoza@chuk.rw (CW); mowiens@outlook.com (MOW)

## Abstract

Post-discharge death is a key contributor to pediatric mortality in sub-Saharan Africa. To address this period's morbidity and mortality, evidence is needed to inform resource prioritization and policy development. No studies have been conducted in Rwanda, limiting understanding of post-discharge mortality. This study aimed to determine the incidence of and risk factors for post-discharge mortality among children under five admitted with suspected sepsis in Rwanda's rural and urban healthcare settings. We conducted a prospective, epidemiologic cohort study of post-discharge mortality in children ages 0–60 months admitted for suspected or confirmed infection in two Rwandan hospitals, one rural (Ruhengeri) and one urban (Kigali), from May 2022 to February 2023. We collected clinical, laboratory, and sociodemographic data on admission and follow-up vital statistics at 2-, 4-, and 6-months post-discharge. Of 1218 children enrolled, 115 (9.4%) died, with half in-hospital (n = 57, 4.7%) and half post-discharge (n = 58, 4.7%). Post-discharge mortality was lower in the 6–60-month cohort (n = 30, 3.5%) than in the 0–6-month cohort (10%) and higher in Kigali (n = 37, 10.3%) vs. Ruhengeri (n = 21, 2.7%). Median time to post-discharge death was 38 days (IQR: 16-97.5) in the 0–6-month cohort and 33 days (IQR: 12–76) in the 6–60-month cohort. In the 0–6 months' cohort, malnutrition (weight-for-age z-score < -3) increased the odds of post-discharge death (aOR 3.31, 95% CI 1.28-8.04), while higher maternal education was protective (aOR 0.15, 95% CI 0.03-0.85). Significant factors in the 6–60-month cohort included an abnormal Blantyre Coma Scale (aOR 3.28, 95% CI 1.47-7.34), travel time to care >;1 hour (aOR 3.54, 95% CI 1.26-9.93), and referral for higher care (aOR 4.13, 95%

**Data availability statement:** This study involves sensitive clinical data from pediatric patients collected across multiple sites, including Rwanda. The following materials have been made openly available through the Sepsis CoLab Dataverse repository on Borealis: the study protocol, informed consent forms, case report forms, data dictionary, and metadata schema. These can be accessed at the following DOI: https://doi.org/10.5683/SP3/NTNTZX. The primary de-identified dataset is available to qualified researchers upon request. Requests must include a description of the intended use and will be reviewed by the Sepsis CoLab Data Governance Committee and the original investigators. This process is designed to ensure compliance with participant consent, local site agreements, and ethical considerations specific to pediatric populations. All requests should be submitted via email to: Email: sepsiscolab@bcchr.ca.

**Funding:** This work was funded by an Early Career Award from the Thrasher Research Fund (to AH), the University of British Columbia (to MW), and the University of California, San Francisco, Department of Emergency Medicine Global Health Section (to AH). The funders had no role in study design, data collection and analysis, decision to publish, or preparation of the manuscript.

**Competing interests:** The authors have declared that no competing interests exist.

CI 1.05-16.27). Children aged <2 months within the 0–6 month cohort exhibited the highest cumulative mortality risk. Post-discharge mortality among Rwandan children remains a challenge, requiring interventions like caregiver counselling, follow-up visits, and community health worker monitoring to reduce mortality rates.

## Introduction

Pediatric mortality following hospital discharge is often an overlooked aspect of child health in Sub-Saharan Africa (SSA) [1,2]. Despite significant decreases in in-hospital mortality rates in pediatric patients in SSA, the early post-discharge period is an especially vulnerable time marked by an increased risk of death, primarily within the first six months [3–5]. An in-depth understanding of the many complex factors contributing to pediatric post-discharge mortality is a critical first step in the design and implementation of effective targeted interventions to reduce the burden of sepsis and improve outcomes in low-resource settings.

Existing research from low- and middle-income country (LMIC) settings has provided valuable insights into the epidemiology and risk factors associated with post-discharge mortality in children. The multi-county Child Health and Mortality Prevention Surveillance Network (CHAIN) study evaluated the causal structure of post-discharge mortality, outlining its complex nature, as it pertains to social, nutritional, and illness-related vulnerability, and identified the importance of malnutrition in post-discharge mortality [6]. In Uganda, the Smart Discharges studies have emphasized that clinical, socioeconomic, behavioral, and lab-based risk factors present on admission can be used for risk stratification of children to inform a more personalized approach to post-discharge care [5,7]. These risk factors have been used to create predictive algorithms to identify children at the highest risk of post-discharge death, in whom low-cost interventions based on this individual risk can be applied to reduce mortality [5,8].

Like most of sub-Saharan Africa, Rwanda has seen dramatic reductions in child mortality, likely the cumulative result of several different, national-level interventional programs [9]. These include interventions such as the introduction of community health insurance, performance-based pay for providers [10], geographical accessibility improvements [11], health system strengthening partnerships [12], nurse mentorship programs [13], community health workers programs [14], and data-driven quality improvement initiatives [15]. Despite a growing body of evidence describing the risks and burden of post-discharge death in children treated for infections [3], its epidemiology in Rwanda has not yet been evaluated.

While Rwanda has made progress in reducing under-five mortality [16], little is known about what happens to children after hospital discharge. Post-discharge deaths are not routinely tracked, and no prospective data exist in Rwanda to guide follow-up care. This study addresses that gap by focusing on children hospitalized with suspected sepsis—a group known to be at high risk. With the complex interplay of health systems, population-, and individual-level risk factors, Rwanda provides



a unique setting for investigating post-discharge mortality among children [14]. Given Rwanda's existing healthcare infrastructure improvements, a better understanding of disparities in post-discharge mortality may encourage additional system-level improvements. This study aimed to investigate the epidemiology of post-discharge mortality in children admitted with suspected sepsis in Rwanda and to identify the key risk factors to inform both clinical practice and health policy, and ultimately improve child survival after discharge.

## Materials and methods

### Study design and setting

This prospective cohort study was conducted at two Rwandan hospitals. Situated in the Northern Province, Ruhengeri Referral Hospital operates as the primary referral center and the only district hospital in Musanze District, serving a largely rural population with a catchment area nearing 500,000 people. The hospital has four total ICU beds and no pediatric-specific ICU capacity. The second, the University Teaching Hospital of Kigali (CHUK), located in Nyarugenge District, Kigali City, is the largest hospital in the country and serves as Rwanda's primary referral center, with a capacity of 483 beds. It also serves as a teaching hospital and center for clinical research for multiple medical schools and provides technical assistance to the surrounding district hospitals.

### Participant recruitment and selection criteria

We prospectively enrolled a cohort of children ages 0–60 months between May 2022 and February 2023. These groups were stratified into 0–6 months and 0–60 months sub-cohorts, given prior variability in risk predictors and model development specific to these age groups, informed by previous studies [5,8]. Inclusion criteria included: any child within this age group admitted with suspected or confirmed infection, as assessed by the treating clinician. This was left to the discretion of the individual clinician and included any child being admitted with a diagnosis of at least one viral, bacterial, or parasitic infection based on clinical (e.g., tachypnea, hypoxia, fever with clinical diagnosis of pneumonia) or laboratory (e.g., rapid diagnostic test positive confirming diagnosis of malaria) criteria. Our previous research in similar settings in Uganda demonstrated that 90% of children enrolled using these same criteria for admission, with a confirmed or suspected infection based on the treating clinician's assessment, met the International Pediatric Sepsis Consensus Conference criteria for sepsis [5,17]. We excluded children living outside the hospital service area, admitted for short-term observation (less than 24 h), or treated for trauma or non-infectious illness. [5,17]. Written informed consent was obtained from all participants' parents or legal guardians. Children whose parents or caregivers refused to participate were excluded from the study.

### Data collection procedures

Data were collected at admission, discharge, and at 2-, 4-, and 6-months post-discharge. A research nurse collected information on clinical history and evaluation, laboratory findings, and sociodemographic characteristics, which mirrored the methodologies used in a similar study conducted in Uganda, ensuring consistency and comparability across studies [5]. All data collection instruments are accessible via the Smart Discharges study Dataverse [18]. Data were gathered directly at the point of care using encrypted study tablets and subsequently uploaded to a Research Electronic Data Capture (REDCap) system [19]. We used a combination of telephone interviews and home visits by research field officers for follow-up visits. These follow-ups focused on vital status, health-seeking behaviors, and any readmissions.

### Variables and measurements

Clinical information collected included vital signs, anthropometric measurements to determine malnutrition status, basic laboratory tests (such as glucose levels, malaria, and HIV rapid diagnostic tests [RDTs], hematocrit, and lactate), observed clinical signs and symptoms, comorbidities, and healthcare history, including any prior hospital admissions. We evaluated

nutritional status using weight-for-age z-scores based on the World Health Organization (WHO) growth standards [20]. Sociodemographic data included maternal and household details, such as mother's age, education level, HIV status, household size, use of bed nets, proximity to the health facility, and availability of clean drinking water. Information on the child's sex was obtained from medical records. At discharge, the study nurses recorded the discharge status (categorized as routine discharge, referral for higher-level care, or unplanned discharge) and feeding status, which were subjectively assessed as feeding well or poorly. Discharge diagnoses were also retrieved from medical records. Field officers contacted caregivers by telephone at 2-, 4-, and 6-months post-discharge to assess the child's vital status, any instances of seeking medical care after leaving the hospital, and details of any readmissions to any health facility, as reported by the caregiver. In cases where contact was lost, we conducted in-person visits to the child's home in the community to gather this information. Although we did include the WHO 2022 standard verbal autopsy tool questions (adapted for Rwanda) [21] for any participants who died following hospital discharge, in-depth verbal autopsy analysis was outside the scope of this study.

Sample size was calculated assuming a rate of post-discharge death of approximately 8%. A sample size of 1000 would provide a~2% margin of error for the outcome of post-discharge death with approximately 80% power and $p = 0.05$.

### Statistical analysis

We performed descriptive statistics to characterize baseline clinical, social/maternal, and discharge variables, stratified by age cohort. We used medians with interquartile ranges for continuous variables and counts with percentages for categorical variables. Multivariate logistic regression models were used to determine risk factors for post-discharge mortality by estimating the odds ratios adjusted for age, sex, and enrollment site. Post-discharge mortality was treated as a binary outcome, and the enrollment site was included as a fixed effect because only two sites were included. We also examined the secondary outcome, readmission post-discharge, using descriptive statistics. We estimated the cumulative hazard for mortality and readmission after discharge with Kaplan-Meier survival curves at four predefined age strata (0–<2 months, 2–6 months, >6–24 months, and >24–60 months). These age strata were chosen to reflect developmental and immunological stages, align with prior pediatric sepsis literature, and enable comparison with similar studies. We had minimal missing data and addressed these using k-nearest neighbor imputation to ensure the robustness and validity of the results. We conducted all statistical analyses using Stata/MP version 15.0 (StataCorp, College Station, TX, USA), R version 4.1.3, and RStudio version 2022.2.3 (RStudio, Boston, MA, USA).

### Ethical considerations

Several institutional review boards approved the study: The University of California, San Francisco (UCSF) on October 8, 2021 (No. 21–34663); the University of British Columbia (UBC) on January 28, 2022 (No. H21-02795), University of Rwanda on December 30, 2021 (No. 411), and University Teaching Hospital of Kigali on January 14, 2022 (No. 005). Informed consent was obtained from the parents or legal guardians of all participants in Kinyarwanda. This manuscript adheres to the STrengthening the Reporting of OBservational studies in Epidemiology (STROBE) statement for cohort studies [22].

### Inclusivity in global research

Additional information regarding the ethical, cultural, and scientific considerations specific to inclusivity in global research is included in the Supporting Information (S1 Checklist).

### Results

We enrolled 1,218 children over the 9-month study period, of whom 1,161 survived to hospital discharge and 1,127 completed follow-up at 6 months post-discharge. There were 115 (9.4%) deaths, evenly split between in-hospital (n = 57, 4.7%) and the post-discharge period (n = 58, 4.7%) (Fig 1).

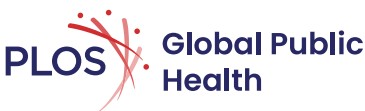

**Fig 1. Study flowchart stratified by age group.**

A flow diagram summarizing participant enrollment, follow-up completion, and outcome ascertainment is presented in Fig 1. The median age of the participants was 13.5 months (IQR 6.1-24.7), and 60% of the cohort was male (n = 676) (S1 Table). Severe malnutrition was common, with 9.1% (n = 103) having a weight-for-age z-score (WAZ) below -3, although this differed significantly between the 0–6-month and 6–60-month age groups (S1 Table and Table 1). Fewer than half of the children (44.4%, n = 500) had a measured fever on presentation (temperature > 37.5 °C), while 18% (n = 208) had measured hypothermia (temperature <36.5 °C). An abnormal Blantyre Coma Scale (BCS) score (≤4) indicating impaired consciousness was observed in 17.2% of patients (n = 194). Only 1.5% (n = 17) were malaria positive, and 0.3% (n = 3) tested positive for HIV. Anemia, defined as a Hemoglobin level <11 g/dL, was present in 36.8% (n = 415) (S1 Table). By disposition, 97.3% of all admitted children were routinely discharged, 1.9% and 0.8% referred to higher care and left against medical advice, respectively (S1 Table).

Characteristics of all enrolled children are detailed in S1 Table, and the characteristics of the two primary age categories are described in Table 1.



**Table 1. Cohort characteristics, disposition, and adjusted odds ratios for post-discharge death, stratified by ages 0-6 months and ages 6-60 months.**

| Variable | 0m to 6m (n=274) | | 6m to 60m (n=853) | |
| --- | --- | --- | --- | --- |
| | N (%)/Median (IQR) | aOR** (95% CI) | N (%)/Median (IQR) | aOR** (95% CI) |
| **Demographics** | | | | |
| **Site, n (%)** | | | | |
| Kigali | 110 (40.2) | reference | 251 (29.4) | reference |
| Ruhengeri | 164 (59.9) | 0.33 (0.15-0.75) | 602 (70.6) | 0.21 (0.10-0.46) |
| **Sex, n (%)** | | | | |
| Female | 112 (40.9) | reference | 339 (39.7) | reference |
| Male | 162 (59.1) | 0.73 (0.33-1.63) | 514 (60.3) | 0.87 (0.41-1.86) |
| Age, months* | 1.4 (0.6-3.6) | 0.91 (0.72-1.14) | 17.9 (11.3-29.8) | 0.98 (0.95-1.01) |
| **Admission anthropometry** | | | | |
| MUAC (mm)*[1] | 120 (104-140) | 0.98 (0.96-1.00) | 150 (140-160) | 0.98 (0.97-1.00) |
| <110/<115 | 83 (30.3) | 2.27 (0.65-7.88) | 26 (3.1) | 2.69 (0.70-10.31) |
| 110-120/115-125 | 70 (25.6) | 3.05 (1.14-8.16) | 42 (4.9) | 2.48 (0.68-9.05) |
| >120/>125 | 121 (44.2) | reference | 785 (92.0) | reference |
| Weight for age z-score | -0.9 (-2.2-0.02) | 0.75 (0.62-0.90) | -0.6 (-1.6-0.3) | 0.61 (0.50-0.74) |
| <-3 | 47 (17.2) | 3.21 (1.28-8.04) | 56 (6.6) | 6.52 (2.63-16.16) |
| -3 to -2 | 24 (8.8) | 2.22 (0.64-7.66) | 92 (10.8) | 3.17 (1.17-8.61) |
| >-2 | 203 (74.1) | reference | 705 (82.7) | reference |
| **Admission clinical assessment** | | | | |
| SpO2, %* | 95 (88-98) | 0.97 (0.92-1.02) | 94 (88-97) | 0.99 (0.95-1.04) |
| Heart rate | 150 (137-162) | 1.02 (1.00-1.04) | 140 (125-154) | 0.99 (0.98-1.01) |
| Respiratory rate* | 46 (40-55) | 1.01 (0.98-1.04) | 40 (34-46) | 1.01 (0.98-1.05) |
| Temperature* | 36.9 (36.5-37.9) | | 37.4 (36.7-38.2) | |
| < 36.5 | 63 (23.0) | 1.01 (0.38-2.71) | 145 (17.0) | 0.40 (0.09-1.82) |
| 36.5-37.5 | 123 (44.9) | reference | 296 (34.7) | Reference |
| >37.5 | 88 (32.1) | 0.53 (0.20-1.43) | 412 (48.3) | 0.83 (0.38-1.80) |
| Abnormal BCS[3]* | 74 (27.0) | 1.05 (0.44-2.54) | 120 (14.1) | 3.28 (1.47-7.34) |
| HIV positive* | 1 (0.4) | – | 2 (0.2) | 11.42 (0.67-194.83) |
| Positive malaria test | 3 (1.1) | – | 14 (1.6) | – |
| Hemoglobin, g/dl* | 12 (10.7-13.5) | 0.83 (0.71-0.97) | 11.3 (10.2-12) | 0.84 (0.71-1.00) |
| No anemia: ≥11g/dL | 192 (70.1) | reference | 520 (61.0) | reference |
| Anemia: <11g/dL | 82 (29.9) | 1.91 (0.83-4.38) | 333 (39.0) | 2.04 (0.93-4.48) |
| Referral | 241 (88.0) | 6.41 (0.82-50.11) | 769 (90.2) | 1.44 (0.46-4.48) |
| Prior antibiotic use | 90 (32.9) | 2.13 (0.73-6.22) | 333 (39.0) | 1.50 (0.58-3.86) |
| Prior antimalarial use | 4 (1.5) | 2.48 (0.23-26.81) | 29 (3.4) | 1.78 (0.48-6.67) |
| Respiratory distress | 66 (24.1) | 1.20 (0.49-2.92) | 162 (19.0) | 1.62 (0.72-3.62) |
| **Maternal and Social Characteristics** | | | | |
| **Time to reach hospital*** | | | | |
| <30m | 112 (40.9) | reference | 341 (40.0) | reference |
| 30 min - 1h | 88 (32.1) | 1.49 (0.55-4.09) | 350 (41.0) | 1.36 (0.48-3.90) |
| >1h | 74 (27.0) | 0.90 (0.31-2.63) | 162 (19.0) | 3.54 (1.26-9.93) |
| **Water source*** | | | | |
| Municipal water/tap | 182 (66.4) | reference | 533 (62.5) | reference |
| Other sources | 92 (33.6) | 0.84 (0.36-1.96) | 320 (37.5) | 2.01 (0.94-4.32) |
| Boil/disinfect/filter water* | 94 (34.3) | 0.21 (0.06-0.72) | 347 (40.7) | 0.74 (0.34-1.62) |

*(Continued)*

**Table 1.** (Continued)

| Variable | 0m to 6m (n = 274) | | 6m to 60m (n = 853) | |
| --- | --- | --- | --- | --- |
| | N (%)/Median (IQR) | aOR** (95% CI) | N (%)/Median (IQR) | aOR** (95% CI) |
| **Maternal education²** | | | | |
| No school or ≤P3 | 33 (12.0) | reference | 118 (13.8) | reference |
| P4 to P6 | 114 (41.6) | 0.47 (0.17-1.31) | 343 (40.2) | 0.41 (0.15-1.09) |
| S1 to S6 | 103 (37.6) | 0.10 (0.03-0.40) | 325 (38.1) | 0.38 (0.14-1.02) |
| > S6 | 23 (8.4) | 0.15 (0.03-0.83) | 65 (7.6) | 0.09 (0.01-0.76) |
| **Discharge Characteristics** | | | | |
| **Discharge status** | | | | |
| Routine discharge | 264 (96.4) | reference | 833 (97.7) | reference |
| Referred to higher level of care | 5 (1.8) | 1.38 (0.14-13.49) | 16 (1.9) | 4.13 (1.05-16.27) |
| Unplanned discharge | 5 (1.8) | 2.40 (0.24-24.39) | 4 (0.5) | – |
| Length of stay | 6 (3-9) | 1.04 (1.01-1.08) | 4 (2-7) | 1.08 (1.05-1.12) |
| **Variables collected only for 0–6-month** | | | | |
| Fontanelle* | 5 (1.8) | 1.73 (0.17-17.19) | – | – |
| Neonatal jaundice* | 15 (5.5) | 2.42 (0.63-9.23) | – | – |
| Sucking well when breastfeeding* | 117 (42.7) | 0.32 (0.13-0.81) | – | – |
| **Duration of present illness*** | | | | |
| <48h | 134 (48.9) | reference | – | – |
| 48h-7d | 104 (38.0) | 2.13 (0.78-5.84) | – | – |
| >7d | 36 (13.1) | 3.57 (1.05-12.18) | – | – |

\* Variables used previously in Smart Discharge prediction models; ¹small number represents cutoff for under 6 months; ²Three participants reported unknown level of education.

3 BCS ≤ 4 was considered abnormal.

**aORs were derived from logistic regression models adjusted for age, sex, and enrolment site.

Abbreviations: aOR = adjusted odds ratio; IQR = interquartile range; BCS = Blantyre Coma scale; HIV = human immunodeficiency virus; $SpO_2$ = oxygen saturation.

## Post-discharge mortality

The overall rate of post-discharge mortality among those discharged alive was 5.2% (n = 58), with a higher cumulative mortality hazard among younger children (**Fig 2**).

Post-discharge deaths occurred at a median of 38 days (IQR 16-97.5) and 33 days (IQR 12–76) in the 0–6-and 6–60 months' groups, respectively, with most deaths occurring in the hospital (57.1% [n = 16] and 70.0% [n = 21], respectively) (**Table 2**).

For the 0–6 months' group, a WAZ below -3 (aOR 3.31, 95% CI 1.28-8.04) was associated with increased risk of mortality, while higher maternal education (aOR 0.15, 95% CI 0.03-0.85) and use of clean drinking water (aOR 0.21, 95% CI 0.06-0.72) were protective (**Table 1**). In the 6–60 months' group, a WAZ below -3 was associated with significantly increased risk of mortality (aOR 6.52, 95% CI 2.63-16.16), along with an abnormal coma score (aOR 3.28, 95% CI 1.47-7.34), travel time over 1 hour to a healthcare facility (aOR 3.54, 95% CI 1.26-9.93), and the need for referral to higher care (aOR 4.13, 95% CI 1.05-16.27). Higher maternal education was also protective in the 6–60 months' group, reducing mortality risk (aOR 0.09, 95% CI 0.01-0.76) (**Table 1**).

## Post-discharge readmission

The rate of post-discharge readmission was 18.9% (n = 213), with 5.8% (n = 65) of children having multiple readmissions at the same facility or other institutions, as reported by caregivers (**Table 2**). The median time to first readmission was

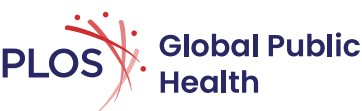

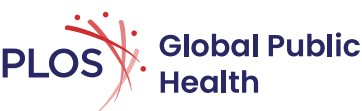

**Fig 2. Hazard curves for post-discharge mortality, stratified by age.**

45 days (IQR 8–125) and 53 days (IQR 25–98) for the 0–6 and 6–60 month groups, respectively. Unlike post-discharge mortality, readmission was not significantly affected by age, although results were trending toward those younger than 2 months having a lower risk of readmission (**Fig 3**).

## Discussion

In our prospective observational study of children under five admitted with suspected or confirmed infections in Rwanda, nearly 1 in 20 children died after discharge, similar to the rate of mortality during the index hospital admission. We found key clinical and socio-behavioral factors are associated with higher odds of post-discharge mortality, including, for both

**Table 2. Secondary endpoint and characteristics of post-discharge deaths.**

| Outcome | 0m to 6m (n=274) | 6m to 60m (n=853) | Total (N=1127) |
|---|---|---|---|
| | N (%)/Median (IQR)* | | |
| **Readmission** | | | |
| Never | 228 (83.2) | 686 (80.4) | 914 (81.1) |
| Once | 36 (13.2) | 112 (13.1) | 148 (13.1) |
| Twice | 8 (2.9) | 35 (4.1) | 43 (3.8) |
| More than twice | 2 (0.7) | 20 (2.3) | 22 (2.0) |
| Number of days from discharge to 1st readmission | 45 (8-125) | 53 (25-98) | 53 (24-108) |
| Number of days from discharge to death | 38 (16-97.5) | 33 (12-76) | 29 (16-90) |
| **Location of death** | | | |
| At home | 8 (28.6) | 8 (26.7) | 16 (27.6) |
| In-transit | 4 (14.3) | 1 (3.3) | 5 (8.6) |
| In hospital | 16 (57.1) | 21 (70.0) | 37 (63.8) |

*IQR = interquartile range.

age groups, severe malnutrition (WAZ<-3); and, for children aged 6–60 months, malnutrition, abnormal coma scores, long travel times to healthcare, and the need for referral to higher-level of care. Infants aged<2 months had the highest risk of death.

These results, as well as the risk factors identified, are largely in line with previous studies conducted in Uganda and elsewhere in East Africa [1–3]. The adverse health effects of malnutrition are well known and include reduced immune competence. [23,24], and deficiencies in macro- and micronutrients [25,26], leading to a cycle of recurrent infections and deteriorating nutritional status, which further increases the risk of post-discharge mortality [27]. We also found higher post-discharge mortality rates in urban Kigali (9%) than in rural Ruhengeri (3%), in contrast to the typical rural-urban health disparity in many LMICs, where patients in rural areas typically fare worse due to limited healthcare access and resources [28]. We hypothesize that elevated mortality rates in Kigali are likely due to CHUK's role as a tertiary care hospital admitting the most critically ill pediatric patients nationwide, including referrals of severe and complex cases from Ruhengeri, which does not have ICU capacity. Unfortunately, baseline mortality rates, particularly post-discharge, were not routinely tracked before our study and are unavailable for comparison. Future exploration of differences among hospital sites' pediatric volume and acuity levels may be valuable for understanding health systems and resource allocation.

Our observation of elevated post-discharge mortality, particularly among young infants and severely malnourished children, aligns with evidence from low- and middle-income countries (LMICs). Studies from Uganda [5] and Bangladesh [29] have identified these vulnerabilities as predictors of mortality after discharge and emphasize the need for ongoing close follow-up of these children. A systematic review across LMIC settings [3]confirms that gaps in post-discharge follow-up care contribute to this increased risk [3]. These findings highlight that inadequate post-discharge health surveillance within resource-constrained health systems may contribute to poor outcomes in high-risk children, emphasizing the need for strategies to improve continuity of care [5].

Rwanda has already implemented programs to improve child health. It is one of the few countries in sub-Saharan Africa to have achieved the Millennium Development Goal (MDG) related to under-5 mortality [30]. With several key health system measures already well established in Rwanda, simple models to identify the "at-risk" child could be leveraged towards practical solutions to address the high post-discharge mortality rates [31]. These include programs such as the well-established Community Health Worker (CHW) program for follow-up care [32], "Mutuelle de Santé," a community-based health insurance program, which lowers financial barriers to potentially improve access to post-discharge services [33],

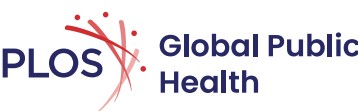

**Fig 3. Hazard curves for first readmission, stratified by age.**

a unified Health Management Information System (HMIS) to facilitate patient tracking and widespread integration of risk-based prediction models [34], and the Mentoring and Enhanced Supervision at Health Centers (MESH) for health-care providers that could be used for training and implementation of system-wide post-discharge care and education packages [35]. These systems argue that more immediate implementation of discharge education and community-based interventions may substantially affect outcomes [36–38]. Other public health campaigns and community-based programs, increased access to quality education, nutritional supplementation, expanded access to healthcare, and improved socio-economic conditions, would also help reduce post-discharge and overall child mortality in Rwanda [39].

This study is the first in Rwanda to examine post-discharge outcomes in pediatric sepsis prospectively. By identifying key risk factors—including age, nutrition, and social vulnerability—we provide locally relevant evidence that may be used to guide discharge planning and targeted follow-up. These findings have practical implications for clinicians, health system leaders, and policymakers working to strengthen continuity of care and reduce preventable deaths after hospital discharge.

## Limitations

The limitations of this study include a small sample from only two hospitals, potentially limiting its generalizability to other regions in Rwanda or similar settings, and a six-month observation period, possibly missing longer-term effects. The primary data collection method and interviews may have introduced recall bias and inaccuracies. Despite this, the study provided detailed information on the severity of the children's conditions and comorbidities. The study did not fully explore all socioeconomic and environmental factors, healthcare quality, disease severity, concurrent illnesses, genetic influences, and healthcare-seeking behaviors affecting post-discharge mortality. We recognize the absence of a systematic approach to tracking diagnoses during readmissions as a limitation of our study. Future research should address these limitations by using a larger, more diverse sample, extending the follow-up period, and conducting a comprehensive analysis of the relevant factors.

## Conclusions

This study highlights that nearly half of all deaths among children hospitalized with suspected sepsis in Rwanda occur after discharge, with the highest risks seen in infants under 6 months, severely malnourished children, and those from socioeconomically vulnerable backgrounds. These findings call for urgent action to strengthen post-discharge follow-up and interventions, especially during the critical first 30–45 days. Examples of high-impact actions that can be taken at the time of hospital discharge may include routine incorporation of nutritional assessments and linkage to malnutrition programs, structured follow-up protocols for vulnerable populations (e.g., neurologically impaired, infants under 6 months), and coordination of subsequent visits with community-based health centers, particularly for patients living long distances from or with difficulty access to referral hospitals.

Potential strategies to reduce this burden are multifactorial and require a coordinated effort by facility and community health workers, policy makers, and non-governmental organizations. Specific needs include (i) the development and deployment of risk models to identify high risk children, (ii) the incorporation of recommendations on scheduled follow-up visits during the early post-discharge period among vulnerable children into national clinical guidelines, and (iii) new multilateral partnerships to develop a national strategy to address the facility to community transition of care for children following discharge to reduce the high burden of post-discharge mortality and to offer a model of care for similar resource-limited settings. This study also calls for more longitudinal research to identify additional factors influencing post-discharge mortality and the development of interventions and implementation strategies for use in low-resource, real-world settings.

The findings underscore the need for a call for action to strengthen health systems in Rwanda and similar sub-Saharan African contexts. Key priorities include integrating discharge planning protocols, providing early follow-up for high-risk pediatric patients, and enhancing community health worker involvement in post-discharge monitoring. Together with broader socioeconomic initiatives, these measures can reduce preventable mortality and improve pediatric survival rates after discharge.

## Supporting information

**S1 Checklist. Supporting Information.** Inclusivity in global research.
(PDF)

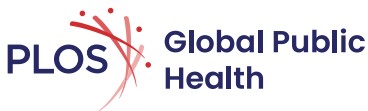

**S1 Table. Supplementary Table 1.** Overall demographics and cohort characteristics. (DOCX)

## Acknowledgments

We would like to acknowledge all past and present members of the Smart Discharges Research program for their efforts in data collection, administration, logistics support, and all study activities, including but not limited to: Godefroid Rucinga, Esperance Umulisa, Didas Mugambinumwe, Jeanne d'Arc Mazimpaka, Claudine Uwingabiye, Theogene Bizimungu, Juliette Unyuzumutima, Peter Lewis, and Martina Knappett.

## Author contributions

**Conceptualization:** Christian Umuhoza, Anneka Hooft, Jessica Trawin, Cynthia Mfuranziza, Emmanuel Uwiragiye, Vuong Nguyen, Nathan Kenya Mugisha, J Mark Ansermino, Matthew O. Wiens.

**Data curation:** Christian Umuhoza, Jessica Trawin, Vuong Nguyen, Matthew O. Wiens.

**Formal analysis:** Christian Umuhoza, Anneka Hooft, Cherri Zhang, Vuong Nguyen, Matthew O. Wiens.

**Funding acquisition:** Anneka Hooft, Jessica Trawin, Aaron Kornblith, Matthew O. Wiens.

**Investigation:** Christian Umuhoza, Anneka Hooft, Matthew O. Wiens.

**Methodology:** Christian Umuhoza, Anneka Hooft, Cherri Zhang, Jessica Trawin, Vuong Nguyen, Aaron Kornblith, J Mark Ansermino, Matthew O. Wiens.

**Project administration:** Christian Umuhoza, Anneka Hooft, Jessica Trawin, Cynthia Mfuranziza, Emmanuel Uwiragiye, Matthew O. Wiens.

**Resources:** Jessica Trawin, Cynthia Mfuranziza, Nathan Kenya Mugisha.

**Supervision:** Christian Umuhoza, Anneka Hooft, Cynthia Mfuranziza, Emmanuel Uwiragiye, Aaron Kornblith, Nathan Kenya Mugisha, J Mark Ansermino, Matthew O. Wiens.

**Validation:** Cherri Zhang.

**Visualization:** Anneka Hooft, Matthew O. Wiens.

**Writing – original draft:** Christian Umuhoza, Anneka Hooft, Matthew O. Wiens.

**Writing – review & editing:** Christian Umuhoza, Anneka Hooft, Cherri Zhang, Jessica Trawin, Vuong Nguyen, Aaron Kornblith, Nathan Kenya Mugisha, J Mark Ansermino, Matthew O. Wiens.

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
