## [Decision Letter · Decision Letter 0]

27 Mar 2025

PGPH-D-24-02667

Post-discharge mortality in suspected pediatric sepsis: insights from rural and urban healthcare settings in Rwanda

Dear Dr. Umuhoza,

Thank you for submitting your manuscript to PLOS Global Public Health. After careful consideration, we feel that it has merit but does not fully meet PLOS Global Public Health’s publication criteria as it currently stands. Therefore, we invite you to submit a revised version of the manuscript that addresses the points raised during the review process.

The manuscript has been evaluated by two reviewers, and their comments are available below. Could you please carefully revise the manuscript to address all comments raised?

We look forward to receiving your revised manuscript.

Kind regards,

Jianhong Zhou

Staff Editor

Journal Requirements:

Additional Editor Comments (if provided):

Reviewers' comments:

Reviewer's Responses to Questions

**Comments to the Author**

1. Does this manuscript meet PLOS Global Public Health’s publication criteria?

Reviewer #1: Yes

Reviewer #2: Yes

2. Has the statistical analysis been performed appropriately and rigorously?

Reviewer #1: Yes

Reviewer #2: Yes

3. Have the authors made all data underlying the findings in their manuscript fully available (please refer to the Data Availability Statement at the start of the manuscript PDF file)?

Reviewer #1: Yes

Reviewer #2: Yes

4. Is the manuscript presented in an intelligible fashion and written in standard English?

Reviewer #1: Yes

Reviewer #2: Yes

Reviewer #1: The journal format and technical production system, as well as the ethics of scientific research, must be adhered to. In addition, I want to know if the mortality rate was due to malnutrition and was this the main reason?

Reviewer #2: Formatting of Data Access Statement:

Ensure the font size, style, and alignment of the Data Access Statement match the required format of the journal.

The placement of this statement should follow the journal’s guidelines—typically under a separate section titled "Data Availability" or "Data Access."

Since the citation style (e.g., "[1]" and "[2]") is correct, no changes are needed there. However, ensure that references are formatted consistently in the reference list.

2. Issues with the Conclusion:

The conclusion does not fully align with the study’s objectives and findings.

It should directly reflect the study’s key results and not introduce new ideas that were not analyzed.

The authors should explicitly link their findings to the research objectives to maintain coherence.

3. Actionable Recommendations Needed:

The conclusion should not only summarize findings—it must include specific, actionable recommendations based on the study’s results.

Examples of improvements:

Strengthening post-discharge monitoring programs for at-risk pediatric patients.

Implementing targeted nutritional support interventions for children with malnutrition.

Developing policies to improve healthcare access in socially vulnerable populations.

The recommendations should clearly state who (healthcare providers, policymakers, NGOs, etc.) should implement them and how they can be applied in practice.

4. Justification of the Study:

The rationale for conducting the study should be more explicitly stated in the introduction and discussion sections.

The authors should clearly explain:

Why this research was necessary (e.g., gaps in knowledge, limited data in Rwanda, policy implications).

How their findings contribute to existing literature and why this study is significant.

What impact the study has on clinical practice, policy, or future research directions.

**Do you want your identity to be public for this peer review?** For information about this choice, including consent withdrawal, please see our Privacy Policy

Reviewer #1: No

Reviewer #2: **Yes: ** EMMANUEL ABU BONSRA

---

## [Decision Letter · Decision Letter 1]

21 May 2025

PGPH-D-24-02667R1

Post-discharge mortality in suspected pediatric sepsis: insights from rural and urban healthcare settings in Rwanda

Dear Christian,

Thank you for submitting your manuscript to PLOS Global Public Health. After careful consideration, we feel that it has merit but does not fully meet PLOS Global Public Health’s publication criteria as it currently stands. Therefore, we invite you to submit a revised version of the manuscript that addresses the points raised during the review process.

We look forward to receiving your revised manuscript.

Kind regards,

Collins Otieno Asweto, PhD

Academic Editor

Journal Requirements:

Additional Editor Comments (if provided):

Reviewers' comments:

Reviewer's Responses to Questions

**Comments to the Author**

Reviewer #1: All comments have been addressed

Reviewer #2: All comments have been addressed

publication criteria?

Reviewer #1: Yes

Reviewer #2: No

3. Has the statistical analysis been performed appropriately and rigorously?

Reviewer #1: Yes

Reviewer #2: No

4. Have the authors made all data underlying the findings in their manuscript fully available (please refer to the Data Availability Statement at the start of the manuscript PDF file)?

Reviewer #1: Yes

Reviewer #2: Yes

5. Is the manuscript presented in an intelligible fashion and written in standard English?

Reviewer #1: Yes

Reviewer #2: Yes

Reviewer #1: All things are good

Reviewer #2: Abstract

the abstract lacks keywords and it lacks objective as well. the main objective is missing in the abstract.

introduction

line 64 and 65 lacks intext citation. Authors should kindly re-write their introduction again to suit their topic. there is no gap found in the introduction. Authors should let their reader to know the gaps they identified.

Methods

the methods lacks details element including study population, sample size determinants, inclusion and exclusion and sample method

result

Table 1: Cohort characteristics and disposition, stratified by ages 0-6 months and

211 ages 6-60 months.

what are the authors reporting, regression ? descriptive ? please check the interpretation.

**Do you want your identity to be public for this peer review?** For information about this choice, including consent withdrawal, please see our Privacy Policy

Reviewer #1: **Yes: ** ‪murtadha abbas‬‏

Reviewer #2: No

---

## [Decision Letter · Decision Letter 2]

10 Aug 2025

PGPH-D-24-02667R2

Post-discharge mortality in suspected pediatric sepsis: insights from rural and urban healthcare settings in Rwanda

Dear Umuhoza,

Thank you for submitting your manuscript to PLOS Global Public Health. After careful consideration, we feel that it has merit but does not fully meet PLOS Global Public Health’s publication criteria as it currently stands. Therefore, we invite you to submit a revised version of the manuscript that addresses the points raised during the review process.

We look forward to receiving your revised manuscript.

Kind regards,

Collins Otieno Asweto, PhD

Academic Editor

Journal Requirements:

Reviewers' comments:

Reviewer's Responses to Questions

**Comments to the Author**

Reviewer #1: All comments have been addressed

Reviewer #3: All comments have been addressed

publication criteria?

Reviewer #1: Yes

Reviewer #3: Yes

3. Has the statistical analysis been performed appropriately and rigorously?

Reviewer #1: Yes

Reviewer #3: Yes

4. Have the authors made all data underlying the findings in their manuscript fully available (please refer to the Data Availability Statement at the start of the manuscript PDF file)?

Reviewer #1: Yes

Reviewer #3: No

5. Is the manuscript presented in an intelligible fashion and written in standard English?

Reviewer #1: Yes

Reviewer #3: Yes

Reviewer #1: Akl thinks are good

Reviewer #3: Thank you for the opportunity to review this important and timely research that addresses a critical and under-researched topic.The topic is highly relevant to sub-Saharan Africa, and the lack of prior studies in Rwanda makes this contribution both novel and valuable for informing national healthcare priorities. I Thank the authors for their efforts in conducting a prospective, multisite study and for highlighting the rural-urban disparities in outcomes in Rwanda.

General:

Please improve some grammar and a concise in objective narrative of the manuscript.

Abstract:

Title and Abstract Consistency:

The title refers specifically to "suspected pediatric sepsis", but the abstract does not clearly define how sepsis was identified or suspected. Clarifying this would help contextualize the findings.

While it is understood that post-discharge causes of death can be difficult to verify, the abstract would benefit from the inclusion of any available information on probable or leading cause-specific mortality, even if based on caregiver reports or verbal autopsy methods.

Data clarity: Please clarify the group age range being reported on. For example

"Children aged <2 months exhibited the highest cumulative mortality risk."

This is confusing, as the age cohort is 0–6 months. Does this refer to a sub-analysis within the 0–6-month cohort? Clarify or remove if not directly supported by data.

Improvement on conclusion: The conclusion emphasizes the need for targeted post-discharge interventions, but it would be stronger if it briefly mentioned what kinds of interventions might be most feasible or effective in the Rwandan context mostly what ou saw was successful while tring to prevet death.

Manuscript:

Method section

Please improve on how ou defined “suspected infection” was operationalized.

Please specift why specific age bands were chosen for Kaplan-Meier analysis.

What was followed during verbal autopsy? This should be briefly described or cited. Was it validated in rwandan context? Is it from WHO?

Results section

Consider using a consolidated flow diagram or summary table (e.g., CONSORT-style) for clarity.

Clarify whether readmissions were to the same facility or any healthcare facility, and whether diagnoses at readmission were tracked.

Discussion

The findings are summarized well; however, the discussion section could benefit from more literature explaining why certain factors presented in the results may contribute to post-discharge mortality, including the noted gap in follow-up visits. Is this gap unique to Rwanda, or is it also observed in other LMICs?

This research could inform policymakers and pediatricians in establishing effective discharge planning. Please Consider adding a call to action for health system strengthening in similar settings within Rwanda and across sub-Saharan Africa.

**Do you want your identity to be public for this peer review?** For information about this choice, including consent withdrawal, please see our Privacy Policy

Reviewer #1: No

Reviewer #3: No

---

## [Decision Letter · Decision Letter 3]

5 Oct 2025

PGPH-D-24-02667R3

Post-discharge mortality in suspected pediatric sepsis: insights from rural and urban healthcare settings in Rwanda

Dear Umuhoza,

Thank you for submitting your manuscript to PLOS Global Public Health. After careful consideration, we feel that it has merit but does not fully meet PLOS Global Public Health’s publication criteria as it currently stands. Therefore, we invite you to submit a revised version of the manuscript that addresses the points raised during the review process.

We look forward to receiving your revised manuscript.

Kind regards,

Collins Otieno Asweto, PhD

Academic Editor

Journal Requirements:

Reviewers' comments:

Reviewer's Responses to Questions

**Comments to the Author**

Reviewer #4: All comments have been addressed

Reviewer #5: All comments have been addressed

publication criteria?

Reviewer #4: Yes

Reviewer #5: Partly

3. Has the statistical analysis been performed appropriately and rigorously?

Reviewer #4: Yes

Reviewer #5: Yes

4. Have the authors made all data underlying the findings in their manuscript fully available (please refer to the Data Availability Statement at the start of the manuscript PDF file)?

Reviewer #4: Yes

Reviewer #5: Yes

5. Is the manuscript presented in an intelligible fashion and written in standard English?

Reviewer #4: Yes

Reviewer #5: Yes

Reviewer #4: Reviewers of the original manuscript gave excellent suggestions for improvement to the initial manuscript. The authors respectfully acknowledged the shortcomings of the manuscript and made honest, and in my opinion, successful, efforts to improve the manuscript per the reviewers' suggestions.

The manuscript now reads more clearly, with observations cleaner and more quantitative for explaining post-discharge mortality of children hospitalized for suspected infection.

The authors improved their Discussion by going beyond just the observational aspect of the study and including potential--and doable--actions that healthcare providers, hospital leaders, community, and government can execute to reduce post-discharge mortality.

I commend the reviewers for finding the shortcomings in this manuscript, and the authors for honestly addressing all substantive comments for improvement and for converting their manuscript from an intriguing though average observational study into a stronger observational study with specific calls to action for important solutions to improve health care in this part of sub-Saharan Africa.

I support a recommendation of publication following the changes made in this revised manuscript.

Reviewer #5: Thank you for the opportunity to review your revised manuscript on post-discharge mortality in Rwandan children with suspected sepsis. This is an important and well-conducted study that addresses a critical knowledge gap in Rwanda. The prospective cohort design and comparison between rural and urban sites are notable strengths, and the findings have significant implications for pediatric care in Rwanda and similar settings. The manuscript has been substantially improved following previous revisions; however, several key points require attention to further strengthen the paper for publication.

Major revisions

Methods (Lines 128-129 and 168-169): A major point requiring significant clarification is the methodology surrounding the cause of death and the definition of the study population. There appears to be a critical contradiction regarding the use of verbal autopsies. Your response to previous feedback indicated that verbal autopsies were not performed and that this statement was corrected; however, the current manuscript draft states on lines 168-169 that "we performed verbal autopsies to determine the likely cause of death." This discrepancy must be resolved to ensure the integrity of the reported methods. Similarly, the definition for "suspected sepsis" on lines 128-129 is noted as being based on the treating clinician's assessment. The manuscript would benefit from a more detailed operational definition or a description of the common clinical signs that guided this assessment, as this is fundamental to understanding the cohort's characteristics and ensuring replicability. In addition, the Blantyre Coma Scale has a score please define what abnormal is for clarity.

Conclusions (Lines 331-351): The conclusion section could be more tightly aligned with the specific, statistically significant risk factors identified in your results. While the current recommendations for risk models and follow-up visits are sound, they are somewhat general. The manuscript's impact would be greater if it more explicitly linked the high odds ratios for factors like severe malnutrition (WAZ < -3), abnormal Blantyre Coma Scale, and travel time (>1 hour) to your proposed strategies. For instance, the findings strongly suggest that discharge planning should include mandatory nutritional screening with referrals for support, and that follow-up protocols should be systematically intensified for children with neurological impairment at admission or those living far from health facilities.

Minor revisions

Discussion (Lines 282-288): Your hypothesis regarding the higher mortality at the urban tertiary hospital is plausible and well-articulated. This point could be strengthened by briefly discussing whether any baseline data on illness severity (e.g., admission severity scores) differed significantly between the two sites. If this data is not available, explicitly stating this and framing it as an important area for future investigation would add further nuance. Finally, a thorough proofread for minor grammatical errors would improve the overall readability of the text. Addressing these points will enhance the clarity, consistency, and impact of your valuable research.

Table alignment and formatting: align your variables to the left instead of centralised, eg demographics etc. then align the values on the table to the centre.

**Do you want your identity to be public for this peer review?** For information about this choice, including consent withdrawal, please see our Privacy Policy

Reviewer #4: **Yes: ** Paul S. Eder

Reviewer #5: **Yes: ** Suleiman Idris Ahmad

---

## [Decision Letter · Decision Letter 4]

20 Nov 2025

Post-discharge mortality in suspected pediatric sepsis: insights from rural and urban healthcare settings in Rwanda

PGPH-D-24-02667R4

Dear Umuhoza,

We are pleased to inform you that your manuscript 'Post-discharge mortality in suspected pediatric sepsis: insights from rural and urban healthcare settings in Rwanda' has been provisionally accepted for publication in PLOS Global Public Health.

Best regards,

Julia Robinson

Executive Editor

Reviewer Comments (if any, and for reference):

Reviewer's Responses to Questions

**Comments to the Author**

Reviewer #4: All comments have been addressed

publication criteria?

Reviewer #4: Yes

3. Has the statistical analysis been performed appropriately and rigorously?

Reviewer #4: Yes

4. Have the authors made all data underlying the findings in their manuscript fully available (please refer to the Data Availability Statement at the start of the manuscript PDF file)?

Reviewer #4: Yes

5. Is the manuscript presented in an intelligible fashion and written in standard English?

Reviewer #4: Yes

Reviewer #4: In my opinion, the authors adequately addressed my and other reviewers' concerns with updated text in the manuscript, in particular in the Abstract, Conclusion, and Discussion sections.

**Do you want your identity to be public for this peer review?** For information about this choice, including consent withdrawal, please see our Privacy Policy

Reviewer #4: **Yes: ** Paul S. Eder, Ph.D.
